# Post-translational thioamidation of methyl-coenzyme M reductase, a key enzyme in methanogenic and methanotrophic Archaea

Dipti D Nayak[1], Nilkamal Mahanta[2], Douglas A Mitchell[1,2,3]*, William W Metcalf[1,3]*

[1]Carl R. Woese Institute for Genomic Biology, University of Illinois, Urbana, United States; [2]Department of Chemistry, University of Illinois, Urbana, United States; [3]Department of Microbiology, University of Illinois, Urbana, United States

**Abstract** Methyl-coenzyme M reductase (MCR), found in strictly anaerobic methanogenic and methanotrophic archaea, catalyzes the reversible production and consumption of the potent greenhouse gas methane. The $\alpha$ subunit of MCR (McrA) contains several unusual post-translational modifications, including a rare thioamidation of glycine. Based on the presumed function of homologous genes involved in the biosynthesis of thioviridamide, a thioamide-containing natural product, we hypothesized that the archaeal *tfuA* and *ycaO* genes would be responsible for post-translational installation of thioglycine into McrA. Mass spectrometric characterization of McrA from the methanogenic archaeon *Methanosarcina acetivorans* lacking *tfuA* and/or *ycaO* revealed the presence of glycine, rather than thioglycine, supporting this hypothesis. Phenotypic characterization of the $\Delta ycaO$-*tfuA* mutant revealed a severe growth rate defect on substrates with low free energy yields and at elevated temperatures (39°C - 45°C). Our analyses support a role for thioglycine in stabilizing the protein secondary structure near the active site.
DOI: https://doi.org/10.7554/eLife.29218.001

*For correspondence:
douglasm@illinois.edu (DAM);
metcalf@illinois.edu (WWM)

**Competing interests:** The authors declare that no competing interests exist.

## Introduction

Methyl-coenzyme M reductase (MCR) is a unique enzyme found exclusively in anaerobic archaea, where it catalyzes the reversible conversion of methyl-coenzyme M (CoM, 2-methylmercaptoethane-sulfonate) and coenzyme B (CoB, 7-thioheptanoylthreoninephosphate) to methane and a CoB-CoM heterodisulfide (*Ermler et al., 1997*; *Scheller et al., 2010*):

$$CH_3 - S - CoM + HS - CoB \rightleftharpoons CH_4 + CoM - S - S - CoB$$

This reaction, which has been proposed to proceed via an unprecedented methyl radical intermediate (*Wongnate et al., 2016*), plays a critical role in the global carbon cycle (*Thauer et al., 2008*). In the forward direction, MCR catalyzes the formation of methane in methane-producing archaea (methanogens). In the reverse direction, MCR catalyzes the consumption of methane in methanotrophic archaea (known as ANMEs, for anaerobic oxidation of methane). Together, these processes produce and consume gigatons of methane each year, helping to establish the steady-state atmospheric levels of an important greenhouse gas. MCR displays an $\alpha_2\beta_2\gamma_2$ protein domain architecture and contains two molecules of $F_{430}$, a nickel porphinoid cofactor (*Ermler et al., 1997*; *Zheng et al., 2016*; *Moore et al., 2017*). The reduced Ni(I) form of $F_{430}$ is essential for catalysis (*Goubeaud et al., 1997*), but is highly sensitive to oxidative inactivation, a feature that renders biochemical

characterization of MCR especially challenging. As a result, many attributes of this important enzyme remain uncharacterized.

An unusual feature of MCR is the presence of several modified amino acids within the active site of the α-subunit. Among these are a group of methylated amino acids, including 3-methylhistidine (or $N^1$-methylhistidine), S-methylcysteine, 5(S)-methylarginine, and 2(S)-methylglutamine, which are presumed to be installed post-translationally by S-adenosylmethionine-dependent methyltransferases (*Selmer et al., 2000*; *Kahnt et al., 2007*). 3-methylhistidine is found in all MCRs examined to date, whereas the presence of the other methylated amino acids is variable among methane-metabolizing archaea (*Kahnt et al., 2007*). A didehydroaspartate modification is also observed in some, but not all, methanogens (*Wagner et al., 2016*). Lastly, a highly unusual thioglycine modification, in which the peptide amide bond is converted to a thioamide, is present in all methanogens that have been analyzed to date (*Figure 1—figure supplement 1*) (*Ermler et al., 1997*; *Kahnt et al., 2007*).

The function(s) of the modified amino acids in MCR has/have not yet been experimentally addressed; however, several theories have been postulated. The 3-methylhistidine may serve to position the imidazole ring that coordinates CoB. This methylation also alters the p$K_a$ of histidine in a manner that would promote tighter CoB-binding (*Grabarse et al., 2000*). The variable occurrence of the other methylated amino acids suggests that they are unlikely to be directly involved in catalysis. Rather, it has been hypothesized that they tune enzyme activity by altering the hydrophobicity and flexibility of the active site pocket (*Kahnt et al., 2007*; *Grabarse et al., 2000*). A similar argument has been made for the didehydroaspartate residue (*Wagner et al., 2016*). In contrast, three distinct mechanistic hypotheses have implicated the thioglycine residue in catalysis. One proposal is that the thioglycine facilitates deprotonation of CoB by reducing the p$K_a$ of the sulfhydryl group (*Grabarse et al., 2001*). A second proposal suggests that thioglycine could serve as an intermediate electron carrier for oxidation of a proposed heterodisulfide anion radical intermediate (*Horng et al., 2001*). Lastly, cis-trans isomerization of the thioamide during catalysis could potentially play an important role in coupling the two active sites via a previously proposed two-stroke mechanism for the enzyme (*Goenrich et al., 2005*).

Thioamides are rare in biology. While cycasthioamide is plant-derived (*Pan et al., 1997*), most other naturally occurring thioamides are of bacterial origin. Among these are the ribosomally synthesized and post-translationally modified (RiPP) peptide natural products thioviridamide (*Hayakawa et al., 2006*) and methanobactin (*Kim et al., 2004*; *Kenney and Rosenzweig, 2012*; *Kenney and Rosenzweig, 2013*), as well as closthioamide, which appears to be an unusual non-ribosomal peptide (*Lincke et al., 2010*; *Chiriac et al., 2015*). The nucleotide derivatives thiouridine and thioguanine (*Coyne et al., 2014*), and two additional natural product antibiotics, thiopeptin and Sch 18640 (*Puar et al., 1981*; *Hensens and Albers-Schönberg, 1983*), also contain thioamide moieties. Although the mechanism of thioamide installation in peptides has yet to be established, the identification of the thioviridamide biosynthetic gene cluster provides clues to their origin (*Izawa et al., 2013*). Two of the proteins encoded by this gene cluster have plausible roles in thioamide synthesis. The first, TvaH, is a member of the YcaO superfamily, while the second, TvaI, is annotated as a 'TfuA-like' protein (*Izawa et al., 2013*). Biochemical analyses of YcaO-family proteins indicate that they catalyze the ATP-dependent cyclodehydration of Cys, Ser, and Thr residues to the corresponding thiazoline, oxazoline, and methyloxazoline (*Figure 1*). Many 'azoline' heterocycles undergo dehydrogenation to the corresponding azole, which are prominent moieties in various RiPP natural products classes, such as linear azol(in)e-containing peptides, thiopeptides, and cyanobactins (*Dunbar et al., 2012*; *Dunbar et al., 2014*; *Arnison et al., 2013*; *Burkhart et al., 2017*). The YcaO-dependent synthesis of peptidic azol(in)e heterocycles often requires a partner protein, which typically is the neighboring gene in the biosynthetic gene cluster (*Burkhart et al., 2015*; *Dunbar et al., 2015*; *Wright et al., 2006*). Based on enzymatic similarity, the TfuA protein encoded adjacent to the YcaO in the thioviridamide pathway may also enhance the thioamidation reaction, perhaps by recruiting a sulfurtransferase protein. Although not RiPPs, thiouridine and thioguanine biosynthesis requires the use of sulfurtransferases (*Coyne et al., 2013*; *Palenchar et al., 2000*).

The biosynthesis of the thioglycine in MCR was proposed to occur by a mechanism similar to that used to produce thioamide-containing natural products (*Kahnt et al., 2007*), a prediction made six years prior to the discovery of the thioviridamide biosynthetic gene cluster (*Izawa et al., 2013*). Given their putative role in thioamidation reactions, it is notable that YcaO homologs were found to be universally present in an early analysis of methanogen genomes, resulting in their designation as

**Figure 1.** Comparison of reactions catalyzed by YcaO proteins. *Top,* Biochemically characterized YcaO proteins involved in the synthesis of azol(in)e-containing ribosomal natural products catalyze the ATP-dependent cyclodehydration of cysteine, serine, and threonine residues. Shown is the conversion of peptidic cysteine to thiazoline. This reaction proceeds via an *O*-phosphorylated hemiorthoamide intermediate. *Bottom,* An analogous reaction is believed to occur in the biosynthesis of the thioamide bond in thioviridamide. Rather than an adjacent cysteine acting as the nucleophile, an exogenous source of sulfide ($S^{2-}$) is required for this reaction.
DOI: https://doi.org/10.7554/eLife.29218.002
The following figure supplement is available for figure 1:

**Figure supplement 1.** A view of the MCR active site using the crystal structure of *M. barkeri* (Protein DataBank entry 1E6Y).
DOI: https://doi.org/10.7554/eLife.29218.003

'methanogenesis marker protein 1' (*Basu et al., 2011*). Genes encoding TfuA homologs are also ubiquitous in methanogens, usually encoded in the same locus as *ycaO*, similar to their arrangement in the thioviridamide biosynthetic gene cluster. Therefore, both biochemical and bioinformatic evidence are consistent with these genes being involved in thioglycine formation. In this report, we use the genetically tractable methanogen *Methanosarcina acetivorans* to show that installation of the thioamide bond at Gly$_{465}$ in the α-subunit of MCR requires the *ycaO-tfuA* locus (*Figure 1—figure supplement 1*). As MCR is essential for growth and survival (*Rother et al., 2005*), the viability of *ycaO-tfuA* mutants precludes the hypothesis that the thioamide residue is essential for MCR activity. Instead, our phenotypic analyses support a role for thioglycine in maintaining a proper structural conformation of residues near the active site, such that absence of this modification in MCR might be particularly detrimental on growth substrates that have low free energy yields as well as under additionally destabilizing conditions such as elevated temperatures.

## Results

### Phylogenetic analyses of TfuA and YcaO in methanogenic and methanotrophic archaea

To examine the possibility that YcaO and TfuA are involved in McrA thioamidation, we reassessed the distribution, diversity, and phylogeny of genes encoding these proteins in sequenced microbial genomes, which today comprise a much larger dataset than when 'methanogenesis marker protein 1' was first proposed (*Basu et al., 2011*). Significantly, all complete methanogen and ANME genome sequences encode a YcaO homolog, with a few strains encoding two copies (*Figure 2*). The YcaO sequences form a distinct, well-supported clade that also includes homologs from ammonia-oxidizing archaea (*Figure 2—figure supplement 1*). Excluding the second YcaO in strains that encode two copies, the YcaO tree is congruent with the phylogeny of methanogens reconstructed using housekeeping or *mcr* genes (*Figure 2*). Thus, it is likely that YcaO was acquired early in the evolution of methanogens and maintained largely through vertical inheritance, as expected for a trait that coevolved with MCR (*Borrel et al., 2013*; *Vanwonterghem et al., 2016*).

TfuA homologs are encoded in the overwhelming majority of MCR-encoding taxa, although a few species lack this gene. The latter include *Methanopyrus kandleri* and *Methanocaldococcus janaschii*, two species previously shown to contain the thioglycine modification (*Kahnt et al., 2007*) (*Figure 2*). Thus, TfuA cannot be obligatorily required for thioglycine installation. The archaeal TfuA proteins

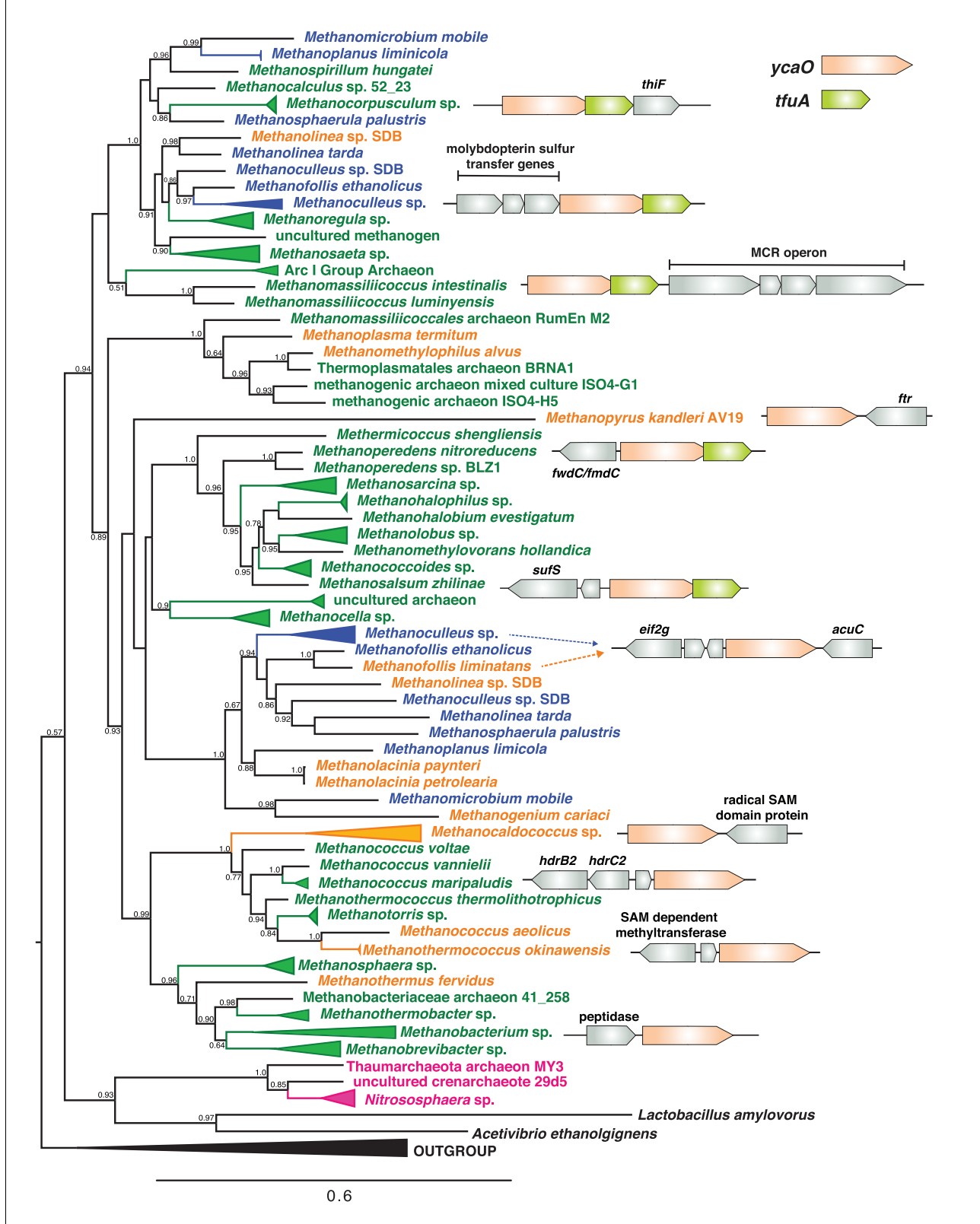

**Figure 2.** A maximum-likelihood phylogenetic tree of YcaO homologs in archaea. Taxa in green, blue, and orange depict methanogens and anaerobic methane-oxidizing archaea (ANMEs) with sequenced genomes. Taxa shown in green contain a single copy of *ycaO* and *tfuA*, those in blue contain two copies of *ycaO* and one copy of *tfuA*, while those in orange contain one copy of *ycaO*, but do not encode *tfuA*. Taxa shown in pink are other archaea that also encode *ycaO* and *tfuA* homologs. The gene neighborhoods of selected taxa are also depicted, showing the common co-localization with

*Figure 2 continued on next page*

*Figure 2 continued*

genes annotated as having a role in sulfur metabolism or methanogenesis. The node labels indicate support values calculated using the Shiomdaira-Hasegawa test using 1000 resamples. Support values less than 0.5 have not been shown. The outgroup displayed derives from a variety of bacterial YcaO proteins.

DOI: https://doi.org/10.7554/eLife.29218.004

The following figure supplements are available for figure 2:

**Figure supplement 1.** (Panel A) Unrooted maximum-likelihood phylogeny of 1,000 YcaO sequences retrieved from the NCBI non-redundant protein sequence database using the corresponding sequence from *Methanosarcina acetivorans* as a search query.

DOI: https://doi.org/10.7554/eLife.29218.005

**Figure supplement 2.** Unrooted maximum-likelihood phylogeny of 1000 TfuA sequences retrieved from the NCBI non-redundant protein sequence database using the corresponding sequence from *Methanosarcina acetivorans* as a search query.

DOI: https://doi.org/10.7554/eLife.29218.006

also fall within a single clade that is largely congruent with the evolutionary history of archaea (*Figure 2—figure supplements 1–2*); however, unlike YcaO, the methanogen-containing clade includes numerous bacterial homologs (*Figure 2—figure supplement 2*).

The genomic context of *tfuA* and *ycaO* in methanogens supports a shared or related function, perhaps involving sulfur incorporation and/or methanogenesis (*Figure 2*). When both are present, the two genes comprise a single locus in which the stop codon of the upstream *ycaO* gene overlaps with the start codon of the downstream *tfuA* gene, suggesting that they are co-transcribed. In several instances, additional genes involved in sulfur metabolism such as *thiF*, *sufS*, as well as *moaD*, *moaE* and *moeB* (involved in molybdopterin biosynthesis) (*Zhang et al., 2010*; *Park et al., 2003*), are found in the genomic vicinity. Occasionally, genes encoding enzymes involved in methanogenesis, including the *Methanomassiliicoccus* MCR operon, are locally encoded (*Figure 2*).

## TfuA and YcaO are not essential in *Methanosarcina acetivorans*

To test their role in thioglycine installation, we generated a mutant lacking the *ycaO-tfuA* locus in the genetically tractable methanogen *Methanosarcina acetivorans*. Based on the hypothesis that thioglycine may be imperative for MCR activity, and the knowledge that the MCR-encoding operon (*mcrBCDAG*) is essential (*Guss et al., 2008*), we first examined the viability of Δ*ycaO-tfuA* mutants using a recently developed Cas9-based assay for gene essentiality (*Nayak and Metcalf, 2017*). This assay compares the number of transformants obtained using a Cas9 gene-editing plasmid with and without a repair template that creates a gene deletion. Essential genes give similar, low numbers of transformants with or without the repair template; whereas non-essential genes give *ca.* $10^3$-fold higher numbers with the repair template. Our data demonstrate that deletion of the *ycaO-tfuA* locus has no impact on viability (*Figure 3*). Several independent mutants were clonally purified and verified by PCR prior to phenotypic characterization (*Figure 3—figure supplement 1*).

## The Δ*ycaO-tfuA* mutant lacks the McrA Gly$_{465}$ thioamide

To test whether YcaO and TfuA are involved in the post-translational modification of Gly$_{465}$ in McrA, we isolated the McrA protein from cell extracts of the isogenic parent (hereafter referred to as wild-type or WT), the Δ*ycaO-tfuA* mutant, as well as from Δ*tfuA* and Δ*ycaO* individual mutants (*Figure 3—figure supplement 2*). After performing SDS-PAGE, the appropriate Coomassie-stained bands were excised and then subjected to in-gel trypsin digestion. The resulting peptides were then analyzed by matrix-assisted laser desorption-ionization time-of-flight mass spectrometry (MALDI-TOF-MS). The mass spectrum from the WT strain showed a peak at *m/z* 3432 Da (*Figure 4*) corresponding to the peptide ($_{461}$LGFF**G**FDLQDQCGATNVLSYQGDEGLPDELR$_{491}$) with Gly$_{465}$ being thioamidated (*Kahnt et al., 2007*; *Grabarse et al., 2000*). The identity of this peptide was confirmed by high-resolution electrospray ionization tandem MS analysis (HR-ESI-MS/MS, [M + 3H]$^{3+}$ expt. *m/z* 1144.8608 Da; calc. *m/z* 1144.8546 Da; 5.4 ppm error; *Figure 4—figure supplement 1A*). This peptide also contains the recently reported didehydroaspartate modification at Asp$_{470}$ (*Wagner et al., 2016*) and S-methylation at Cys$_{472}$ (*Selmer et al., 2000*). Consistent with the involvement of TfuA-associated YcaO proteins in thioamide formation, we noted the absence of the *m/z* 3432 Da species in the mass spectrum of similarly treated Δ*ycaO-tfuA*, Δ*tfuA*, and Δ*ycaO* samples. Instead, a predominant *m/z* 3416 Da species appeared, which was 16 Da lighter, consistent with replacement of sulfur by

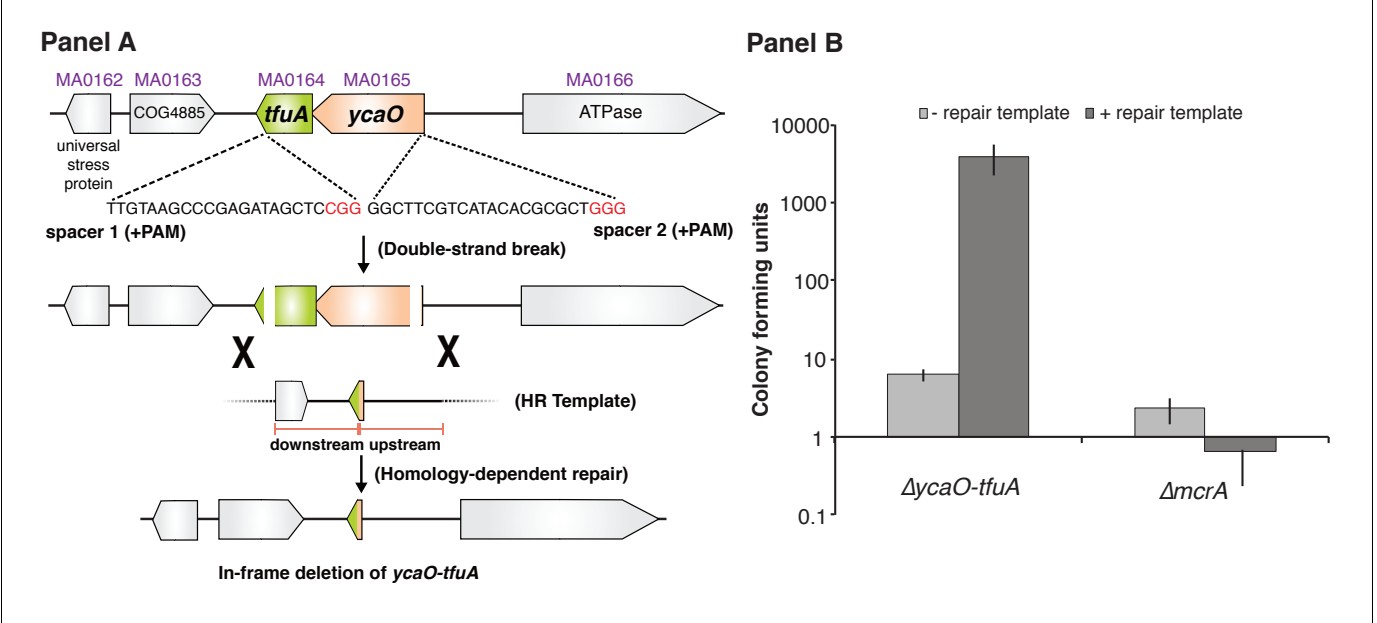

**Figure 3.** Panel A, Experimental strategy for deletion of the *ycaO-tfuA* locus in *M. acetivorans* using a Cas9-mediated genome editing technique. Co-expression of the Cas9 protein along with a single guide RNA (sgRNA) with a 20 bp target sequence flanked by a 3' NGG protospacer adjacent motif on the chromosome (PAM; in red) generates double-stranded breaks at the *ycaO-tfuA* locus. Repair of this break via homologous recombination with an appropriate repair template (HR Template) generates an in-frame deletion on the host chromosome. Panel B, A Cas9-based assay for gene essentiality. Mean transformation efficiencies of plasmids with (dark gray) or without (light gray) an appropriate repair template are shown. The known essential gene *mcrA* is included as a positive control. The error bars represent one standard deviation of three independent transformations.

DOI: https://doi.org/10.7554/eLife.29218.007

The following figure supplements are available for figure 3:

**Figure supplement 1.** A PCR-based screen to genotype the *ycaO-tfuA* locus in *Methanosarcina acetivorans*.

DOI: https://doi.org/10.7554/eLife.29218.008

**Figure supplement 2.** The experimental strategy for generating an in-frame deletion of either the *tfuA* locus (Panel A) or the *ycaO* locus (Panel B) in *M. acetivorans* using a Cas9-mediated genome editing technique.

DOI: https://doi.org/10.7554/eLife.29218.009

oxygen (*Figure 4*). HR-ESI-MS/MS analysis of the corresponding peptide from the Δ*ycaO-tfuA* mutant confirmed the identity of this peptide as being McrA Leu$_{461}$-Arg$_{491}$ ([M + 3H]$^{3+}$ expt. *m/z* 1139.5316 Da; calc. *m/z* 1139.5290 Da; 2.3 ppm error; *Figure 4—figure supplement 1B*). During HR-ESI-MS/MS analysis of the WT, thioamide-containing peptide, we did not observe fragmentation between Gly$_{465}$ and Phe$_{466}$. In contrast, fragmentation was seen at this location in the Δ*ycaO-tfuA* strain (b$_5$ ion, *Figure 4—figure supplement 1B*). The lack of fragmentation in the WT peptide was anticipated based on the greater double bond character of C-N bonds in thioamides (*Wiberg and Rablen, 1995*).

To confirm that mutations in *ycaO* and/or *tfuA* were responsible for loss of the thioglycine modification, we transformed the Δ*ycaO-tfuA* mutant with a self-replicating plasmid containing the *ycaO-tfuA* locus fused to the tetracycline-inducible promoter P*mcrB*(*tetO4*) (*Guss et al., 2008*). Upon tryptic digestion of MCR from cultures supplemented with tetracycline, a peak at *m/z* 3432 Da reappeared (*Figure 4*). Similar results were obtained when complementation experiments were performed with the *ycaO* or *tfuA* genes in the corresponding single mutants (*Figure 4*).

The *m/z* 3375 Da species present in all samples corresponds to an unrelated tryptic peptide, McrA Asp$_{392}$-Arg$_{421}$ (*Figure 4—figure supplement 2*). We also observed a peak at *m/z* 1496 in the WT and mutant samples corresponding to a *N*-methylhistidine-containing tryptic peptide (His$_{271}$-Arg$_{284}$, *Figure 4—figure supplement 3A*). Thus, thioglycine formation is not a prerequisite for this modification. Additionally, a peak at *m/z* 1561 was observed, corresponding to McrA Phe$_{408}$-Arg$_{421}$, shows that Gln$_{420}$ remains unmodified in *M. acetivorans*, as previously been observed for *Methanosarcina barkeri* (*Figure 4—figure supplement 3B*) (*Kahnt et al., 2007*).

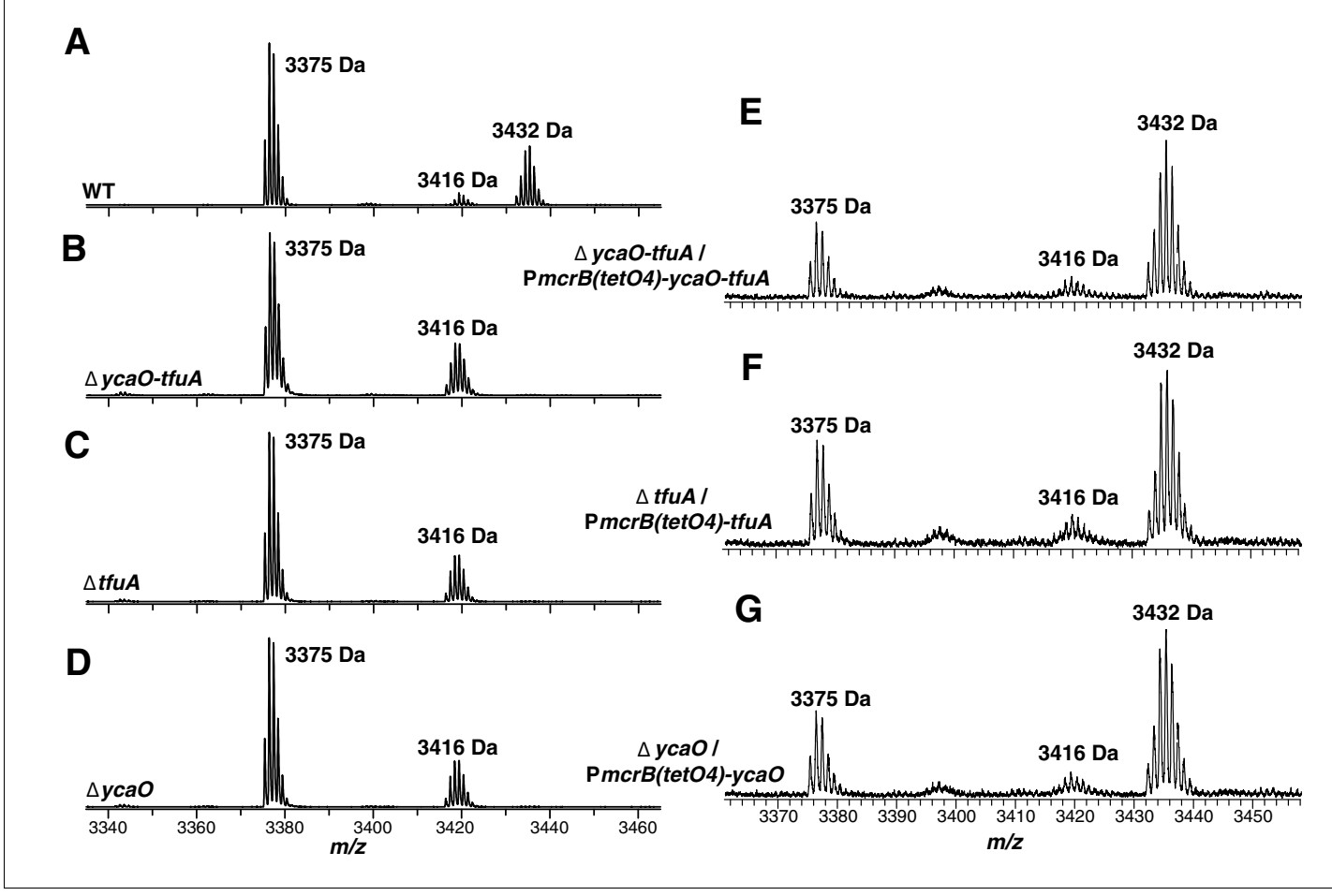

**Figure 4.** MALDI-TOF MS analysis of McrA. Panel A, Spectrum obtained from trypsinolysis of WT MCR, which contains the thioglycine-modified $L_{461}$-$R_{491}$ peptide ($m/z$ 3432 Da). A small amount of the unmodified $L_{461}$-$R_{491}$ peptide ($m/z$ 3416 Da) is also observed, which probably arises via non-enzymatic hydrolysis of the thioamide bond as previously reported (**Kahnt et al., 2007**). Panels B, C, D, Spectrum obtained from trypsinolysis of McrA from the $\Delta ycaO$-$tfuA$, $\Delta tfuA$, $\Delta ycaO$ strains, respectively, containing only the unmodified $L_{461}$-$R_{491}$ peptide ($m/z$ 3416 Da). Panels E, F, G. Spectrum obtained from trypsinolysis of MCR from the $\Delta ycaO$-$tfuA$ mutant complemented with P$mcrB$($tetO4$)-$ycaO$-$tfuA$, $\Delta tfuA$ mutant complemented with P$mcrB$($tetO4$)-$tfuA$, $\Delta ycaO$ mutant complemented with P$mcrB$($tetO4$)-$ycaO$, respectively. Tetracycline was added to a final concentration of 100 µg/mL to the growth medium to induce expression from the P$mcrB$($tetO4$) promoter. All three spectra contain the thioglycine-modified $Leu_{461}$-$Arg_{491}$ peptide ($m/z$ 3432 Da). The $m/z$ 3375 Da species present in all samples corresponds to McrA residues $D_{392}$-$R_{421}$.

DOI: https://doi.org/10.7554/eLife.29218.010

The following figure supplements are available for figure 4:

**Figure supplement 1.** (Panel A) High-resolutions electrospray ionization tandem mass spectrometry (HR-ESI MS/MS) analysis of a wild-type tryptic peptide from *Methanosarcina acetivorans* McrA ($Leu_{461}$-$Arg_{491}$, $m/z$ 3432 Da).

DOI: https://doi.org/10.7554/eLife.29218.011

**Figure supplement 2.** HR-ESI-MS/MS of the tryptic peptide from *Methanosarcina acetivorans* McrA ($Asp_{392}$-$Arg_{421}$, $m/z$ 3375 Da) present in the wild-type and $\Delta ycaO$-$tfuA$, $\Delta tfuA$, $\Delta ycaO$ deletion strains.

DOI: https://doi.org/10.7554/eLife.29218.012

**Figure supplement 3.** (Panel A) Matrix-assisted laser desorption/ionization time-of-flight mass spectrometry (MALDI-TOF MS) analysis of the wild-type (WT, top) and $\Delta ycaO$-$tfuA$ deletion strain (bottom) spectra identify the McrA tryptic peptide, $His_{271}$-$Arg_{284}$ ($m/z$ 1496 Da), containing the known *N*-methylhistidine modification.

DOI: https://doi.org/10.7554/eLife.29218.013

## Growth defects associated with loss of the McrA thioglycine modification

To understand potential phenotypic consequences of losing the thioglycine modification, we quantified the doubling time and growth yield by monitoring changes in optical density during growth of

**Table 1.** Growth phenotypes of *M. acetivorans* WT and the Δ*ycaO-tfuA* mutant on different substrates at 36°C.

| Substrate* | Strain | Doubling time (h)† | p-value | Maximum OD$_{600}$† | p-value |
|---|---|---|---|---|---|
| Methanol | WT | 6.94 ± 0.41 | 0.87 | 5.35 ± 0.31 | 0.26 |
| | Δ*ycaO-tfuA* | 6.89 ± 0.15 | | 5.06 ± 0.39 | |
| TMA | WT | 10.92 ± 0.54 | 0.84 | 9.57 ± 0.67 | 0.31 |
| | Δ*ycaO-tfuA* | 10.98 ± 0.43 | | 8.91 ± 0.73 | |
| Acetate | WT | 54.23 ± 4.03 | 0.0002 | 0.44 ± 0.01 | 0.0005 |
| | Δ*ycaO-tfuA* | 124.33 ± 12.52 | | 0.35 ± 0.02 | |
| DMS | WT | 31.27 ± 1.28 | <0.0001 | 0.60 ± 0.01 | <0.0001 |
| | Δ*ycaO-tfuA* | 58.28 ± 1.45 | | 0.44 ± 0.01 | |

*TMA, trimethylamine; DMS, dimethylsulfide.

†The error represents the 95% confidence interval of the mean for three biological replicates and the p-values were calculated using two-tailed unpaired t-tests.

DOI: https://doi.org/10.7554/eLife.29218.017

*M. acetivorans* on a variety of substrates (*Table 1*). While no significant differences were observed during growth on methanol or trimethylamine (TMA) at 36°C, the Δ*ycaO-tfuA* mutant had substantially longer generation times and lower cell yields on both dimethylsulfide (DMS) and acetate (*Table 1*). Interestingly, the growth phenotype on methanol medium was strongly temperature-dependent, with no observed differences at 29°C and 36°C, but severe defects observed for the Δ*ycaO-tfuA* mutant at 39°C and 42°C. Furthermore, unlike WT, the mutant was incapable of growth at 45°C (*Figure 5*).

## Thioglycine does not influence the global stability of MCR

Given that the C-N bond rotation barrier is higher for thioamides relative to amide (*Wiberg and Rablen, 1995*), we tested if the presence of thioglycine promotes the thermal stability of MCR. We evaluated thermal stability by assaying the melting temperature of MCR purified from WT and the Δ*ycaO-tfuA* mutant. A tandem affinity purification (TAP) tag consisting of a 3 × FLAG tag and a Twin-Strep tag was introduced immediately upstream of the start codon of *mcrG* (*Figure 5—figure supplement 1A*). The proteins were affinity-purified using a Strep-tactin resin under aerobic conditions (*Figure 5—figure supplement 1B*). Using the SYPRO Orange-based Thermofluor assay (*Huynh and Partch, 2015*), the melting temperature of WT MCR was 67.3 ± 0.3°C (mean ± 95% confidence interval of three technical replicates) whereas the melting temperature for the MCR variant lacking thioglycine was 69.1 ± 0.2°C. Additionally, quantification of the protein and F$_{430}$ concentrations by Bradford assay and absorption at 280/430 nm demonstrated that the F$_{430}$ content was not significantly affected by the presence of thioglycine (*Figure 5—figure supplement 2*).

## Discussion

The loss of the thioglycine modification in the Δ*ycaO, ΔtfuA, and ΔycaO-tfuA* mutants shows that both of these genes are required for the thioamidation of McrA. Although this could be an indirect requirement, we believe it is more likely that the YcaO/TfuA proteins directly catalyze modification of McrA. This conclusion is mechanistically compatible with biochemical analyses of YcaO homologs. YcaO enzymes that carry out the ATP-dependent cyclodehydration of beta-nucleophile-containing amino acids have been extensively investigated (*Burkhart et al., 2017*). Such cyclodehydratases coordinate the nucleophilic attack of the Cys, Ser, and Thr side chain on the preceding amide carbonyl carbon in a fashion reminiscent of intein splicing (*Perler et al., 1997*) (*Figure 1*). The enzyme then *O*-phosphorylates the resulting oxyanion and subsequently *N*-deprotonates the hemiorthoamide, yielding an azoline heterocycle. An analogous reaction can be drawn for the YcaO-dependent formation of peptidic thioamides. The only difference is that an exogenous equivalent of sulfide is required for the thioamidation reaction, rather than an adjacent beta-nucleophile-containing amino acid for azoline formation (*Figure 1*).

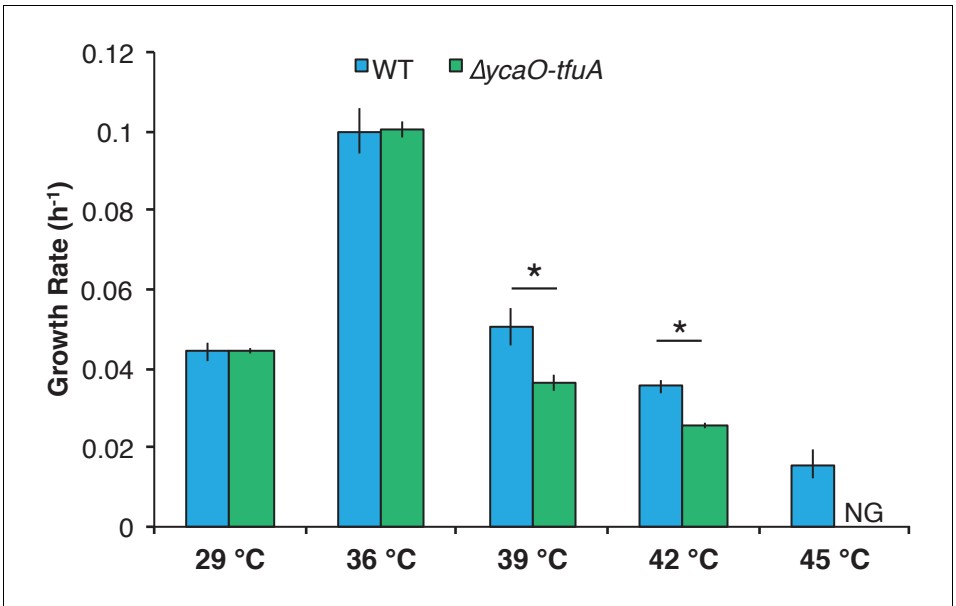

**Figure 5.** Growth rate of WT (blue) and the *ΔycaO-tfuA* mutant (green) in HS medium with 125 mM methanol at the indicated temperatures. A statistically significant difference (p<0.01) using a two-tailed unpaired t-test is indicated with an *. NG indicates that no growth was detected after one month of incubation.
DOI: https://doi.org/10.7554/eLife.29218.014

The following figure supplements are available for figure 5:

**Figure supplement 1.** (Panel A) Nucleotide and amino acid sequence of the N-terminal TAP (tandem affinity purification) tag inserted immediately upstream of the start codon of *mcrG* (locus tag, MA4547) using the Cas9-mediated genome editing technique.
DOI: https://doi.org/10.7554/eLife.29218.015

**Figure supplement 2.** UV-Vis absorbance spectra of tagged-MCR purified from the WT strain (blue) and the *ΔycaO-tfuA* mutant (orange).
DOI: https://doi.org/10.7554/eLife.29218.016

---

Most YcaO cyclodehydratases require a partner protein for catalytic activity. The earliest investigated YcaO partner proteins are homologs of the ThiF/MoeB family, which are related to E1 ubiquitin-activating enzymes (*Dunbar et al., 2014*; *Schulman et al., 2009*). These YcaO partner proteins, as well as the more recently characterized 'ocin-ThiF' variety (*Dunbar et al., 2015*), contain a ~ 90 residue domain referred to as the RiPP precursor peptide Recognition Element (RRE), which facilitates substrate recognition by interacting with the leader peptide. Considering these traits of azoline-forming YcaOs, it is possible that thioamide-forming YcaOs require the TfuA partner to facilitate binding to the peptidic substrate. Alternatively, TfuA may recruit and deliver sulfide equivalents by a direct or indirect mechanism. In this regard, it is noteworthy that the *ycaO-tfuA* locus can be found adjacent to genes involved in sulfur and molybdoterin metabolism (*Figure 2*). Many of these genes encode proteins with rhodanese-like homology domains, which are well-established sulfurtransferases. These enzymes typically carry sulfur in the form of a cysteine persulfide, a non-toxic but reactive equivalent of $H_2S$ (*Palenchar et al., 2000*; *Matthies et al., 2005*). Akin to rare cases of azoline-forming, partner-independent YcaOs, certain methanogens lack a bioinformatically identifiable TfuA (e.g. *Methanopyrus kandleri* and *Methanocaldococcus* sp., *Figure 2*). Whether these YcaOs act independently or use an as yet-unidentified partner protein remains to be seen. Clearly, further in vitro experimentation will be required to delineate the precise role of TfuA in the thioamidation reaction.

The viability of the *ΔycaO-tfuA* mutant raises significant questions as to the role of thioglycine in the native MCR enzyme, especially considering its universal presence in all MCRs examined to date. We considered three hypotheses to explain this result. First, we examined the possibility that thioglycine modification is involved in enhancing the reaction rate (*Kahnt et al., 2007*; *Horng et al.,*

*2001*). Although it has not been explicitly determined, MCR is thought to catalyze the rate-limiting step of methanogenesis (*Scheller et al., 2010*; *Wongnate et al., 2016*). Therefore, the absence of thioglycine might lead to a corresponding decrease in the growth rate, with more pronounced defects on substrates that lead to the fastest growth. However, we observed the opposite: the most pronounced defects were observed with growth substrates that support the slowest WT growth rates (*Table 1*). Next, we considered the possibility that the thioglycine influences substrate affinity. $C_1$ units enter methanogenesis at the level of $N^5$-methyl-tetrahydrosarcinapterin (CH$_3$-H$_4$SPt) during growth on acetate (*Galagan et al., 2002*; *Deppenmeier et al., 1999*), but at the level of methyl-CoM during growth on DMS, methanol, and TMA (*Figure 6*) (*Bose et al., 2008*; *Fu and Metcalf, 2015*). Significantly, these entry points are separated by an energy-dependent step that is coupled to production or consumption of the trans-membrane $Na^+$ gradient. As a result, intracellular levels of methyl-CoM, CoM, and CoB could possibly be significantly different depending on the entry point into the methanogenic pathway. Because we observed growth defects on substrates that enter at both points (i.e. DMS and acetate), we suspect that the growth deficiency phenotype is unlikely to be related to changes in substrate affinity. A third explanation we considered was that thioglycine increases the stability of MCR. In this model, unstable MCR protein would need to be replaced more often, creating a significant metabolic burden for the mutant. Consistent with our results, this additional burden would be exacerbated on lower energy substrates like DMS and acetate, especially given that MCR comprises ~10% of the total protein (*Rospert et al., 1990*). Further, one would expect that a protein stability phenotype would be exaggerated at higher temperatures, which we observed during growth on methanol (*Figure 5*). Thus, multiple lines of evidence support the idea that the growth-associated phenotypes stemming from the deletion of TfuA and YcaO are caused by decreased MCR stability. However, the melting temperature of MCR from the Δ*ycao-tfuA* mutant was modestly higher than that purified from the WT. As the thioglycine is buried deep within the active site of MCR (*Ermler et al., 1997*), these data are consistent with the notion that thioamidation does not affect the global stability of the protein. Instead, it suggests that the thioglycine modification impacts the local stability in the vicinity of the buried active site of MCR (*Figure 1—figure supplement 1*), which in turn might have a detrimental effect under catalytic turnover conditions.

The chemical properties of amides relative to thioamides are consistent with our hypothesis. Although amides tend to be planar, their rotational barriers are lower than for the corresponding thioamide and thus they are more conformationally flexible. This is especially true for glycine.

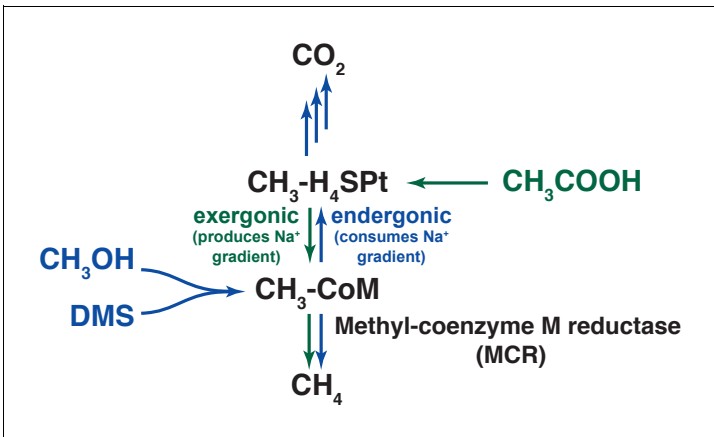

**Figure 6.** An overview of methanogenic metabolism in *M. acetivorans*. Methylotrophic substrates such as methanol (CH$_3$OH) or dimethylsulfide (DMS, CH$_3$-S-CH$_3$) enter the methanogenic pathway via *S*-methylation of coenzyme M (CoM) and subsequently disproportionate to methane (CH$_4$) and carbon dioxide (CO$_2$; metabolic flux is shown as blue arrows). Notably, the first step in oxidation of CH$_3$-CoM to CO$_2$ is the energy-requiring transfer of the methyl moiety to generate methyl-tetrahydrosarcinapterin (CH$_3$-H$_4$SPt). In contrast, acetic acid (CH$_3$COOH) enters the pathway at the CH$_3$-H$_4$SPt level, followed by reduction to CH$_4$ (green arrows). Thus, the second step of the pathway is exergonic.

DOI: https://doi.org/10.7554/eLife.29218.018

Considering the negligible electronegativity difference between sulfur and carbon, the thioamide carbonyl is not polarized like an amide carbonyl (*Wiberg and Rablen, 1995*). Further, sulfur has a larger van der Waals radius than oxygen resulting in a thioamide bond length that is ~40% longer than the amide bond (1.71 Å versus 1.23 Å) (*Petersson et al., 2014*), which presents additional steric hindrances to backbone flexibility. Finally, the p$K_a$ of thioamides is lower than amides, making thio-glycine a stronger hydrogen bond donor than glycine (*Lee et al., 2002*), which again could reduce conformational flexibility. Taken together, it is chemically reasonable to state that the increased flexi-bility of the unmodified glycine in the Δ*ycaO-tfuA* mutant would render the contorted conformation of the peptide backbone in the Gly$_{462}$-Leu$_{469}$ region of McrA considerably less stable (*Figure 1—fig-ure supplement 1*). Indeed, this region adopts a rather strange configuration, with the side chain of Phe$_{463}$ bending back towards the thioamide of Gly$_{465}$. The molecular alignment and distance (3.5 Å) suggests a potential π-cation interaction between these residues(*Figure 1—figure supplement 1*) The thioamide sulfur is also optimally distanced (3.6 Å) and geometrically positioned to form an H-bond with the side chain of Asn$_{501}$ and the backbone N-H of Leu$_{469}$. There also appears to be a hydrophobic interaction between the thioamide and the beta carbon of Leu$_{469}$, which would be energetically unfavorable with a more polarized amide. These observations suggest that the local stability in the vicinity of the active site imparted by the thioamide is required for protein stability and increased half-life of the MCR protein. A conclusive test of this hypothesis will require compre-hensive structural and biochemical characterization of the active MCR variant lacking the thioglycine modification, which is beyond the scope of this work.

Finally, the temperature-sensitive phenotype of the Δ*ycaO-tfuA* mutant has potential implications regarding the evolution and ecology of methanogenic archaea. Based on this result, it seems reason-able to speculate that the thioglycine modification would be indispensable for thermophilic metha-nogens. It is often posited that the ancestor of modern methanogens was a thermophilic organism (*Gribaldo and Brochier-Armanet, 2006*; *Forterre, 2015*; *Weiss et al., 2016*; *López-García et al., 2015*). If so, one would expect the thioglycine modification to be present in most methanogenic lin-eages, being stochastically lost due to genetic drift only in lineages that grow at low temperatures where the modification is not required. In contrast, if methanogenesis evolved in a cooler environ-ment, one might expect the distribution of the modification to be restricted to thermophilic line-ages. Thus, the universal presence of the thioglycine modification supports the thermophilic ancestry of methanogenesis. Indeed, the *ycaO-tfuA* locus is conserved even in *Methanococcoides burtonii*, a psychrophilic methanogen isolated from Ace Lake in Antarctica, where the ambient temperature is always below 2°C (*Franzmann et al., 1992*). It will be interesting to see whether this modification is maintained by other methanogenic and methanotrophic archaea growing in low temperature environments.

## Materials and methods

### Bioinformatics analyses

The 1000 closest homologs were extracted from the NCBI non-redundant protein database using the YcaO amino acid sequence (MA0165) or the TfuA amino acid sequence (MA0164) as queries in BLAST-P searches. The amino acid sequences of these proteins were aligned using the MUSCLE plug-in (*Edgar, 2004*) with default parameters in Geneious version R9 (*Kearse et al., 2012*). Approx-imate maximum-likelihood trees were generated using FastTree version 2.1.3 SSE3 using the Jones-Taylor-Thornton (JTT) model +CAT approximation with 20 rate categories. Branch support was cal-culated using the Shimodaira-Hasegawa (SH) test with 1000 resamples. Trees were displayed using Fig Tree v1.4.3 (http://tree.bio.ed.ac.uk/software/figtree/).

### Strains, media, and growth conditions

All *M. acetivorans* strains were grown in single-cell morphology (*Sowers et al., 1993*) in bicarbon-ate-buffered high salt (HS) liquid medium containing 125 mM methanol, 50 mM trimethylamine hydrochloride (TMA), 40 mM sodium acetate, or 20 mM dimethylsulfide (DMS). Cultures were grown in sealed tubes with N$_2$/CO$_2$ (80/20) at 8–10 psi in the headspace. Most substrates were added to the medium prior to autoclaving. DMS was added from an anaerobic stock solution maintained at 4°C immediately prior to inoculation. Growth rate measurements were conducted with three

independent biological replicate cultures acclimated to the energy substrate or temperature as indicated. A 1:10 dilution of a late-exponential phase culture was used as the inoculum for growth rate measurement. Plating on HS medium containing 50 mM TMA solidified with 1.7% agar was conducted in an anaerobic glove chamber (Coy Laboratory Products, Grass Lake, MI) as described previously (*Metcalf et al., 1996*). Solid media plates were incubated in an intra-chamber anaerobic incubator maintained at 37°C with N$_2$/CO$_2$/H$_2$S (79.9/20/0.1) in the headspace as described previously (*Metcalf et al., 1998*). Puromycin (CalBiochem, San Diego, CA) was added to a final concentration of 2 µg/mL from a sterile, anaerobic stock solution to select for transformants containing the *pac* (puromycin transacetylase) cassette. The purine analog 8-aza-2,6-diaminopurine (8ADP) (R. I. Chemicals, San Diego, CA) was added to a final concentration of 20 µg/mL from a sterile, anaerobic stock solution to select against the *hpt* (phosphoribosyltransferase) cassette encoded on pC2A-based plasmids. Anaerobic, sterile stocks of tetracycline hydrochloride in deionized water were prepared fresh shortly before use and added to a final concentration of 100 µg/mL to the growth medium of strains containing *ycaO* and/or *tfuA* fused to P*mcrB*(*tetO4*). *E. coli* strains were grown in LB broth at 37°C with standard antibiotic concentrations. WM4489, a DH10B derivative engineered to control copy-number of oriV-based plasmids (*Kim et al., 2012*), was used as the host strain for all plasmids generated in this study (*Supplementary file 1*). Plasmid copy number was increased dramatically by supplementing the growth medium with sterile rhamnose to a final concentration of 10 mM.

## Plasmids

All plasmids used in this study are listed in *Supplementary file 1*. Plasmids for Cas9-mediated genome editing were designed as described previously (*Nayak and Metcalf, 2017*). For designing the complementation plasmids, the *ycaO* and/or *tfuA* coding sequence(s) were amplified and fused to the P*mcrB*(*tetO4*) promoter in pJK029A (*Guss et al., 2008*) linearized with *NdeI* and *HindIII* by the Gibson assembly method as described previously in *Nayak and Metcalf, 2017*. Standard techniques were used for the isolation and manipulation of plasmid DNA. WM4489 was transformed by electroporation at 1.8 kV using an *E. coli* Gene Pulser (Bio-Rad, Hercules, CA). All pDN201-derived plasmids were verified by Sanger sequencing at the Roy J. Carver Biotechnology Center, University of Illinois at Urbana-Champaign and all pAMG40 cointegrates were verified by restriction endonuclease analysis.

## In silico design of sgRNAs for gene-editing

All target sequences used for Cas9-mediated genome editing in this study are listed in *Supplementary file 2*. Target sequences were chosen using the CRISPR site finder tool in Geneious version R9 (61). The *M. acetivorans* chromosome and the plasmid pC2A were used to score off-target binding sites.

## Transformation of *M. acetivorans*

All *M. acetivorans* strains used in this study are listed in *Supplementary file 3*. Liposome-mediated transformation was used for *M. acetivorans* as described previously (*Metcalf et al., 1997*) using 10 mL of late-exponential phase culture of *M. acetivorans* and 2 µg of plasmid DNA for each transformation.

## In-gel tryptic digest of McrA

Mid-exponential phase cultures of *M. acetivorans* grown in 10 mL HS medium containing 50 mM TMA were harvested by centrifugation (3000 × *g*) for 15 min at 4°C (DuPont, Wilmington, DE). The cell pellet was resuspended in 1 mL lysis buffer (50 mM NH$_4$HCO$_3$, pH = 8.0) and harvested by centrifugation (17,500 × *g*) for 30 min at 4°C (DuPont, Wilmington, DE). An equal volume of the supernatant was mixed with 2 × Laemmli sample buffer (Bio-Rad, Hercules, CA) containing 5% β-mercaptoethanol, incubated in boiling water for 10 min, loaded on a 4–20% gradient Mini-Protean TGX denaturing SDS-PAGE gel (Bio-Rad, Hercules, CA) and run at 70 V until the dye-front reached the bottom of the gel. The gel was stained using the Gel Code Blue stain reagent (Thermo Fisher Scientific, Waltham, MA) as per the manufacturer's instructions. Bands corresponding to McrA (*ca.* 60 kDa) were excised and cut into *ca.* 1 × 1 mm cubes. The gel slices from a single lane were

destained with 50% acetonitrile, the corresponding protein was reduced with 10 mM dithiothreitol (DTT), and digested with 1.5 µg sequencing-grade trypsin (Promega, Madison, WI) at 37°C for 16–20 hr in the presence of 5% (v/v) n-propanol. The digested peptides were extracted and dried as described previously (*Shevchenko et al., 2006*).

## MS analysis of tryptic peptides

MALDI-TOF-MS analysis was performed using a Bruker UltrafleXtreme MALDI TOF-TOF mass spectrometer (Bruker Daltonics, Billerica, MA) in reflector positive mode at the University of Illinois School of Chemical Sciences Mass Spectrometry Laboratory. The samples were desalted using C-18 zip-tips using aqueous acetonitrile and sinapic acid in 70% acetonitrile as the matrix. Data analysis was carried out using the Bruker FlexAnalysis software. For HR-ESI MS/MS, samples were dissolved in 35% aq. acetonitrile and 0.1% formic acid. Samples were directly infused using an Advion TriVersa Nanomate 100 into a ThermoFisher Scientific Orbitrap Fusion ESI-MS. The instrument was calibrated weekly, following the manufacturer's instructions, and tuned daily with Pierce LTQ Velos ESI Positive Ion Calibration Solution (Thermo Fisher Scientific, Waltham, MA). The MS was operated using the following parameters: resolution, 100,000; isolation width (MS/MS), 1 *m/z*; normalized collision energy (MS/MS), 35; activation q value (MS/MS), 0.4; activation time (MS/MS), 30 ms. Data analysis was conducted using the Qualbrowser application of Xcalibur software (Thermo Fisher Scientific, Waltham, MA). HPLC-grade reagents were used to prepare samples for mass spectrometric analyses.

## MCR purification

MCR was purified from 250 mL of mid-exponential phase culture grown in HS +50 mM TMA at 36°C. Protein purification was performed under aerobic conditions and cells were harvested by centrifugation (3000 $\times$ *g*) for 15 min at 4°C. The cell pellet was lysed in 5 mL of Wash buffer (50 mM $NaH_2PO_4$, 300 mM NaCl, pH = 8.0) and the cell lysate was clarified by centrifugation (17,500 $\times$ *g*) for 30 min at 4°C. The supernatant fraction was loaded on a column containing 2 mL Strep-tactin XT Superflow resin (50% suspension; IBA Lifesciences, Goettingen, Germany) equilibrated with 4 mL of the Wash buffer. The column was washed twice with 4 mL of Wash buffer and the purified protein was eluted in four fractions with 0.5 mL Elution buffer (50 mM $NaH_2PO_4$, 300 mM NaCl, 50 mM biotin, pH = 8.0) per fraction. The protein concentration in each fraction was estimated using the Coomassie Plus (Bradford) assay kit (Pierce Biotechnology, Thermo-Scientific, Rockford, IL, USA) with BSA (bovine serum albumin) as the standard per the manufacturer's instructions. The highest protein concentration (*ca.* 600 µg/mL) was obtained in fraction 3 therefore this fraction was used to conduct the Thermofluor assay. To visualize the purified protein, 10 µL of the crude cell extract as well as fraction 3 were mixed with an equal volume of 2 $\times$ Laemmli sample buffer (Bio-Rad, Hercules, CA) containing 5% β-mercaptoethanol, incubated in boiling water for 10 min, loaded on a 4–20% gradient Mini-Protean TGX denaturing SDS-PAGE gel (Bio-Rad, Hercules, CA) and run at 70 V until the dye-front reached the bottom of the gel. The gel was stained using the Gel Code Blue stain reagent (Thermo Fisher Scientific, Waltham, MA) as per the manufacturer's instructions. All three bands corresponding to the MCR subunits were observed and verified by LC-MS analyses of the tryptic peptides from fraction 3 of the purified protein sample at the Protein Sciences facility at the Roy J. Carver Biotechnology Center at the University of Illinois, Urbana-Champaign, USA. A fourth band, identified as a hypothetical protein encoded by MA3997, was also detected in each sample (*Figure 5—figure supplement 1*).

## Thermofluor assay

A 5000 $\times$ concentrate of SYPRO Orange protein gel stain in DMSO (Thermo-Fisher Scientific, Waltham, MA, USA) was diluted with the Elution buffer to generate a 200 $\times$ stock solution. 5 µL of the 200 $\times$ stock solution of SYPRO Orange protein gel stain was added to 45 µL of purified protein with a concentration of *ca.* 600 µg/mL in 96-well optically clear PCR plates. A melt curve was performed on a Mastercycler ep realplex machine (Eppendorf, Hamburg, Germany) using the following protocol: 25°C for 30 s, ramp to 99°C over 30 min, return to 25°C for 30 s. SYPRO Orange fluorescence was detected using the VIC emission channel and the temperature corresponding to the inflection point of the first derivative (dI/dT) was determined to be the melting temperature ($T_m$). Appropriate

no-dye controls and no-protein controls were examined and each sample was run in triplicate. The Thermofluor assay was conducted within 12 hr of protein purification.

## UV-Vis spectrum of MCR

The spectrum was obtained on a Cary 4000 UV-Vis spectrophotometer (Varian, Palo Alto, CA, USA) using 200 µL of protein sample (1.5 µM each) in a quartz supracil cuvette (Hellma Analytics, Mullheim, Germany). The scanning range was 200–800 nm and the Elution buffer was used as a blank.

## Acknowledgements

We thank Graham A Hudson (DAM lab) for technical assistance with the HR and tandem MS data acquisition and Dr. Peter Yau at the Protein Sciences facility at the Roy J Carver Biotechnology Center at the University of Illinois for identifying the protein that co-purified with MCR.

## Additional information

### Funding

| Funder | Grant reference number | Author |
|---|---|---|
| National Institute of General Medical Sciences | GM097142 | Douglas A Mitchell |
| Life Sciences Research Foundation | Simons Foundation | Dipti D Nayak |
| U.S. Department of Energy | DE-FG02-02ER15296 | William W Metcalf |

The funders had no role in study design, data collection and interpretation, or the decision to submit the work for publication.

### Author contributions

Dipti D Nayak, William W Metcalf, Conceptualization, Data curation, Formal analysis, Supervision, Funding acquisition, Investigation, Methodology, Writing—original draft, Project administration, Writing—review and editing; Nilkamal Mahanta, Data curation, Formal analysis, Investigation, Writing—original draft, Writing—review and editing; Douglas A Mitchell, Conceptualization, Supervision, Funding acquisition, Writing—original draft, Project administration, Writing—review and editing

### Author ORCIDs

Dipti D Nayak http://orcid.org/0000-0002-8390-7251
Nilkamal Mahanta http://orcid.org/0000-0001-8901-2531
Douglas A Mitchell http://orcid.org/0000-0002-9564-0953
William W Metcalf http://orcid.org/0000-0002-0182-0671

### Decision letter and Author response

Decision letter https://doi.org/10.7554/eLife.29218.024
Author response https://doi.org/10.7554/eLife.29218.025

## Additional files

### Supplementary files

• Supplementary file 1. Lists of plasmids
DOI: https://doi.org/10.7554/eLife.29218.019

• Supplementary file 2. List of target sequences for Cas9-mediated genome editing
DOI: https://doi.org/10.7554/eLife.29218.020

• Supplementary file 3. List of strains
DOI: https://doi.org/10.7554/eLife.29218.021

• Transparent reporting form

DOI: https://doi.org/10.7554/eLife.29218.022

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
