## [Decision Letter]

[Editors’ note: a previous version of this study was rejected after peer review, but the authors submitted for reconsideration. The first decision letter after peer review is shown below.]

Thank you for submitting your work entitled "Post-translational thioamidation of methyl-coenzyme M reductase, a key enzyme in methanogenic and methanotrophic Archaea" for consideration by *eLife*. Your article has been reviewed by four peer reviewers and the evaluation has been overseen by a Reviewing Editor and a Senior Editor. The following individuals involved in review of your submission have agreed to reveal their identity: Dianne K Newman (Reviewer #1) and Rudolph Thauer (Reviewer #4).

Our decision has been reached after consultation between the reviewers. Based on these discussions and the individual reviews below, we regret to inform you that your work will not be considered further for publication in *eLife* at this time.

While there is sincere enthusiasm for your work, and a desire to see it ultimately published in *eLife*, the consensus opinion is that, as the manuscript currently stands, the key finding - genetic demonstration that a thioglycine in MCR is installed by YcaO/TfuA and is not essential for catalytic activity - is not a sufficient advance to appeal to the broad *eLife* audience. Though we appreciate that comparing the catalytic activity of WT and mutant MCR is well beyond the scope of the manuscript, some other biochemical follow-up would significantly strengthen other claims of the manuscript (e.g. thermostability, mechanism of YcaO/TfuA, role of TfuA, etc). The evolutionary implications are intriguing, but not yet convincing. For example, why would MCR from *Methanopyrus kandleri* growing above 100°C and psychrophilic methanogens and methanotrophs both have the thioglycine in the active site of MCR conserved? And why would the thioglycine only show up in MCR? If you are able and willing to perform some additional experiments to bolster the claims in the manuscript, we would be happy to reconsider it for *eLife*.

Reviewer #1:

This manuscript presents strong evidence that 2 genes (tfuA and ycaO) are required for installation of a thioglycine into McrA, a critical enzyme in methanogenesis and in organisms that oxidize methane anaerobically. These processes are globally significant and challenging to study, so work that improves our understanding of how they function is attractive for publication in *eLife*.

Positives of the paper include clear writing and well-done genetic and MS experiments. The application of biochemical understanding from the natural product field to identify putitive genes catalyzing installation of a thioglycine in a methanogen is elegant and chemically satisfying. However, while the authors provide physiological data and reasonable logic arguing that the thioglycine confers greater stability to MCR, the evidence supporting this claim is indirect. One concern is that there is some other target of the modification other than McrA that is responsible for the observed phenotype. Perhaps this can be excluded as a possibility, but if so, the authors need to explain why that is. To a naïve reader, this seems like an important missing piece. One option would be to perform a limited set of biochemical experiments to look at McrA stability with and without the thioglycine residue. The authors argue this is beyond the scope of the paper, but is it really? Could this be done by looking for different sized bands of McrA on a western blot (might one predict differential stability would show up in different patterns on a protein gel)? How difficult is it to purify McrA and test for temperature stability for the WT and mutant versions? Alternatively, could a different mutation be introduced that would be predicted to render the McrA protein less thermostable, and could the phenotype be recapitulated? In the absence of biochemical data, that might strengthen the argument. How can the authors be confident that other proteins produced under the experimental conditions are not modified by TfuA/YcaO and responsible for the phenotypes? At a minimum, the authors need to directly address this concern in their discussion.

The Abstract and Discussion end with speculation about the implications of this work for an evolutionary hot start for methanogenesis. This is a bold claim, and the phylogenetic evidence is intriguing, but for this claim to be convincing, it is important to have solid evidence that the thioglycine in McrA is what causes the lack of thermostability. I would like to see this manuscript published in *eLife*, but encourage the authors to perform another experiment to strengthen this critical point.

Reviewer #2:

This paper addresses the biosynthesis and functional significance of a post-translational modification found in methyl-coenzyme M reductase, an enzyme that produces methane in anaerobic archaea and may also be involved in methane oxidation by methanotrophic archaea. Several post-translationally modified residues are present in or near the MCR active site in the McrA subunit, including a thioglycine. Candidate proteins for producing this thioamide include a YcaO homolog paired with a TfuA homolog. YcaOs catalyze formation of thiazolines from cysteines, and TfuA-like proteins are hypothesized to play a role in sulfur delivery and/or substrate recognition. Bioinformatic analysis presented here supports a role for these proteins in McrA thioamidation, and genetic disruption of both genes in *Methanosarcina acetivorans* coupled with mass spectrometric analysis indicates that the thioglycine is not formed in their absence. While the cells are still viable, some differences in growth are observed between the wildtype and *ΔycaO-tfuA* strains. In particular, the knockout strain grows more slowly on DMS and acetate, and more slowly on methanol if the temperature is raised. From these growth data, the authors suggest that the thioglycine modification does not affect MCR catalysis or substrate affinity, but instead increases its thermal stability. Although this model is consistent with the results, biochemical characterization of the MCR is essential to addressing this hypothesis. In addition, forming a thioamide seems like a metabolically expensive way to increase thermostability as many other proteins are thermostable without such post-translational modifications. The authors note that biochemical work is beyond the scope of this study, but I am not sure that the knockout alone merits publication in *eLife*; it seems more suited to a specialized journal.

In terms of thinking about the modification and its potential role in thermal stability or other aspects of function, it would be helpful to have a picture of the MCR active site showing where the glycine residue is located and what interactions it has with other residues.

Reviewer #3:

Metcalf et al. address interesting questions regarding post-translational modifications of the last enzyme of the methanogenic pathway in archaea. It has been known for almost two decades that some residues in the active site of the methyl-coenzyme M reductase (Mcr) are modified. Of particular interest is the thioamidation of glycine. The goal was to identify genes encoding enzymes responsible for this modification. The authors build a case for the analysis of two genes (ycaO, tfuA) that tracked in some cases immediately upstream to the Mcr operon. The authors use *Methanosarcina acetivorans* to perform phenotypic analyses of strains with deletions of the genes of interest. The observed growth phenotypes are explained on the basis of the entry point of different substrates into the methanogenic pathway. The results are consistent with the explanations provided. However, the authors do not validate the presumed biochemical activities of YcaO or TfuA, hence leading to much speculation. The effects reported here could be indirectly leading to the absence of thioamidated glycine. The authors start the Discussion section by stating '[…] these genes (ycaO, tfuA) are directly involved in the post-translational modification of McrA'; I disagree. To support such a statement the authors must show that YcaO/TfuA can modify McrA in vitro.

Reviewer #4:

The manuscript by Metcalf et al. for *eLife* describes the important finding that the post-translational installation of thioglycine in methyl-coenzyme M reductase in the methanogenic archaeon *Methanosarcina acetivorans* is mediated by the tfuA-ycaO locus. This experimental procedure showing this is perfect and the results obtained convincing. The excellent manuscript should be published after considering the following minor points:

Discussion paragraph three and following: Yes, MCR is generally thought to be the rate limiting enzyme in methanogenesis, however, the only step in methanogenesis that is physiologically irreversible, is the heterodisulfide reductase reaction rather than the MCR reaction. Thus MCR is most probably not rate limiting. This should be considered or at least discussed.

Of the many glycins in McrA only one is converted to a thioglycin and this glycine is buried deep within the McrABC complex. The post-translational modification must therefore occur as long as the protein is not yet folded. Generally enzymes mediating post-translational modifications specifically recognize the sequence upstream and downstream of the amino acid to be modified. Could the authors comment on this in the manuscript with a few sentences?

[Editors’ note: what now follows is the decision letter after the authors submitted for further consideration.]

Thank you for resubmitting your work entitled "Post-translational thioamidation of methyl-coenzyme M reductase, a key enzyme in methanogenic and methanotrophic Archaea" for further consideration at *eLife*. Your revised article has been favorably evaluated by Gisela Storz (Senior editor) and four reviewers, one of whom is a member of our Board of Reviewing Editors.

We appreciate your efforts to respond to our feedback by performing new experiments and revising the text accordingly. The manuscript has been improved but there are some remaining issues that need to be addressed before acceptance, as outlined below:

The consensus is that in the absence of supporting biochemical data, the manuscript still pushes a mechanistic message too strongly. Please moderate the discussion. Your results have shown that the thioglycine within the active site is not essential for full activity, not that it lacks a direct catalytic role. In addition, the paragraph on the discussion of growth phenotypes is convoluted and overly speculative. It would be better to tighten and moderate this paragraph, in the absence of supporting data. Growth phenotypes should also be mentioned in the Abstract, so it is clear that the phenotype of the double mutant extends beyond the lack of a PTM. While we agree that additional biochemical experiments are outside the scope of this work, for this manuscript to be published in *eLife* as a genetics/physiology paper, these aspects need to be airtight. Towards that end, please include complementation experiments for the mutants.

With these modifications, we anticipate your second revision would be publishable in *eLife*.

Reviewer #1:

The new experiments and revisions to the text have alleviated my previous concerns and the manuscript is much improved.

While there are many questions remaining that need to be addressed at the biochemical level, this study has made an important new contribution and is done rigorously.

Reviewer #2:

This is a revised version in which the authors have addresses most of the questions and suggestions made by the reviewers. The interpretations remain speculative but the experimental results are too important not to be published.

Reviewer #3:

In the revised version of this manuscript, Metcalf et al. show that strains carrying null alleles of *tfuA* or *ycaO* do not modify the glycine of interest. In addition, results from new experiments show that thioglycine is not needed to stabilize MCR. The new hypothesis is now that the stabilizing effect occurs at the level of the active site; that idea was not tested. In the absence of experimental data, the authors softened the wording regarding the direct involvement of TfuA and YcaO.

Without a question, this is a very interesting problem, but I am not convinced that the revised paper has reached the level of an *eLife* publication.

Reviewer #4:

In this revised manuscript, the authors have added two experiments. First, they have constructed separate knockouts of *ycaO* and *tfuA* as well as the double knockout, and demonstrate by mass spectrometric analysis that both are necessary for thioglycine formation. This is not surprising. Second, they have isolated the MCR produced by the double knockout strain and show that its melting temperature is unchanged, disproving their previous hypothesis regarding thermal stability. They have revised the hypothesis to invoke localized stability at the active site; the overall argument regarding the properties of amides versus thioamides is the same. A stability experiment is one of the approaches the editor suggested as biochemical follow up so in that sense, the authors have done what was requested.

---

## [Author Response]

[Editors’ note: the author responses to the first round of peer review follow.]

*Reviewer #1:*
*This manuscript presents strong evidence that 2 genes (tfuA and ycaO) are required for installation of a thioglycine into McrA, a critical enzyme in methanogenesis and in organisms that oxidize methane anaerobically. These processes are globally significant and challenging to study, so work that improves our understanding of how they function is attractive for publication in eLife.*
*Positives of the paper include clear writing and well-done genetic and MS experiments. The application of biochemical understanding from the natural product field to identify putitive genes catalyzing installation of a thioglycine in a methanogen is elegant and chemically satisfying. However, while the authors provide physiological data and reasonable logic arguing that the thioglycine confers greater stability to MCR, the evidence supporting this claim is indirect. One concern is that there is some other target of the modification other than McrA that is responsible for the observed phenotype. Perhaps this can be excluded as a possibility, but if so, the authors need to explain why that is. To a naïve reader, this seems like an important missing piece. One option would be to perform a limited set of biochemical experiments to look at McrA stability with and without the thioglycine residue. The authors argue this is beyond the scope of the paper, but is it really? Could this be done by looking for different sized bands of McrA on a western blot (might one predict differential stability would show up in different patterns on a protein gel)? How difficult is it to purify McrA and test for temperature stability for the WT and mutant versions? Alternatively, could a different mutation be introduced that would be predicted to render the McrA protein less thermostable, and could the phenotype be recapitulated? In the absence of biochemical data, that might strengthen the argument. How can the authors be confident that other proteins produced under the experimental conditions are not modified by TfuA/YcaO and responsible for the phenotypes? At a minimum, the authors need to directly address this concern in their discussion.*

We thank the reviewer for their insightful critique. We recognize that we had not rigorously shown that the in vivo phenotype resulting from the *ycaO-tfuA* deletion is due to an affect on MCR (rather than on some unidentified target of *ycaO/tfuA*) and we have made a dedicated effort to address this. To do this, we introduced affinity tag at the N-terminus of the *mcrG* CDS in the WT strain as well as in a strain lacking the *ycaO-tfuA* locus in order to aid purification and studies of MCR. (As a side note, this is the first time affinity purification of MCR has been conducted in the native host.) A SYPRO-orange based Thermofluor assay was performed with MCR purified aerobically from each of these strains (Figure 5—figure supplement 1) to determine the melting temperature of the complex containing all three subunits. We did not observe any significant difference in the melting temperature of the MCR complex lacking thioglycine under these conditions. Also, the content of F430 cofactor was unchanged between WT and mutant as determined by UV-Vis spectroscopy.

The outcome of these critical experiments suggests that thioglycine does not affect the global stability of the MCR complex; however, local instability in the region of the MCR active site, which is buried within the α subunit (McrA), cannot be confirmed nor refuted at this time. While we are interested in determining the precise role of the thioglycine, the requisite anaerobic biochemical interrogation of the protein is beyond the scope of the current study.

In addition, a comprehensive proteomic analysis might determine if YcaO/TfuA modify other proteins, but it would be impossible to obtain 100% coverage of all peptides in such an experiment. Therefore, a negative result would still be inconclusive. Given the associated cost and effort required to conduct such an experiment, we feel this is not worth pursuing at this time.

*The Abstract and Discussion end with speculation about the implications of this work for an evolutionary hot start for methanogenesis. This is a bold claim, and the phylogenetic evidence is intriguing, but for this claim to be convincing, it is important to have solid evidence that the thioglycine in McrA is what causes the lack of thermostability. I would like to see this manuscript published in eLife, but encourage the authors to perform another experiment to strengthen this critical point.*

Given the inconclusive data cited above, we have significantly softened this argument in the revised version.

*Reviewer #2:*
*This paper addresses the biosynthesis and functional significance of a post-translational modification found in methyl-coenzyme M reductase, an enzyme that produces methane in anaerobic archaea and may also be involved in methane oxidation by methanotrophic archaea. Several post-translationally modified residues are present in or near the MCR active site in the McrA subunit, including a thioglycine. Candidate proteins for producing this thioamide include a YcaO homolog paired with a TfuA homolog. YcaOs catalyze formation of thiazolines from cysteines, and TfuA-like proteins are hypothesized to play a role in sulfur delivery and/or substrate recognition. Bioinformatic analysis presented here supports a role for these proteins in McrA thioamidation, and genetic disruption of both genes in Methanosarcina acetivorans coupled with mass spectrometric analysis indicates that the thioglycine is not formed in their absence. While the cells are still viable, some differences in growth are observed between the wildtype and ΔycaO-tfuA strains. In particular, the knockout strain grows more slowly on DMS and acetate, and more slowly on methanol if the temperature is raised. From these growth data, the authors suggest that the thioglycine modification does not affect MCR catalysis or substrate affinity, but instead increases its thermal stability. Although this model is consistent with the results, biochemical characterization of the MCR is essential to addressing this hypothesis. In addition, forming a thioamide seems like a metabolically expensive way to increase thermostability as many other proteins are thermostable without such post-translational modifications. The authors note that biochemical work is beyond the scope of this study, but I am not sure that the knockout alone merits publication in eLife; it seems more suited to a specialized journal.*

As noted by other reviewers, and in the letter from the Editor, biochemical characterization of MCR is truly beyond the scope of this manuscript. During the purification process, the Ni(I) in F430 is especially prone to oxidation to Ni(II) and reductive activation of the enzyme in vitro is especially challenging and has only been demonstrated for the MCR from a different, distantly related organism (*Methanothermobacter marburgensis*; Prakash et al. J. Bac 2014). To circumvent this issue, MCR in the cells is oxidized to the stable Ni(III) form in vivo prior to purification, and Ni(III) is reduced to Ni(I) in vitro by the addition of titanium citrate as a reductant. Protocols for the conversion of MCR to the Ni(III) form vary significantly across different methanogenic strains and a robust protocol for *M. acetivorans* has not been developed yet. We hope to conduct such studies in the future and with this paper we have the motivation to do so.

*In terms of thinking about the modification and its potential role in thermal stability or other aspects of function, it would be helpful to have a picture of the MCR active site showing where the glycine residue is located and what interactions it has with other residues.*

We have added a new figure supplement (Figure 1—figure supplement 1) to depict the location of the modified thioglycine residue relative to coenzyme B, coenzyme M, F430, as well as the interaction that it has with other residues in the vicinity.

*Reviewer #3:*
*Metcalf et al. address interesting questions regarding post-translational modifications of the last enzyme of the methanogenic pathway in archaea. It has been known for almost two decades that some residues in the active site of the methyl-coenzyme M reductase (Mcr) are modified. Of particular interest is the thioamidation of glycine. The goal was to identify genes encoding enzymes responsible for this modification. The authors build a case for the analysis of two genes (ycaO, tfuA) that tracked in some cases immediately upstream to the Mcr operon. The authors use Methanosarcina acetivorans to perform phenotypic analyses of strains with deletions of the genes of interest. The observed growth phenotypes are explained on the basis of the entry point of different substrates into the methanogenic pathway. The results are consistent with the explanations provided. However, the authors do not validate the presumed biochemical activities of YcaO or TfuA, hence leading to much speculation. The effects reported here could be indirectly leading to the absence of thioamidated glycine. The authors start the Discussion section by stating '[…] these genes (ycaO, tfuA) are directly involved in the post-translational modification of McrA'; I disagree. To support such a statement the authors must show that YcaO/TfuA can modify McrA* in vitro.

We agree with the reviewer that in vitro experiments would need to be done to prove that YcaO/TfuA directly modify McrA. Thus, we have modified the first sentence of the Discussion section to soften our stance on the direct involvement of TfuA/YcaO in the post-translational installation of thioglycine. However, we have added additional data showing that both proteins are required in vivo. To do this, we have generated single mutants lacking either *ycaO* or *tfuA* and conducted MS analyses of McrA from each of these strains. Our results (subsection “The *∆ycaO-tfuA* mutant lacks the McrA Gly465 thioamide” and Figure 4) indicate that both genes seem to be required for the installation of the thioamide bond.

*Reviewer #4:*
*The manuscript by Metcalf et al. for eLife describes the important finding that the post-translational installation of thioglycine in methyl-coenzyme M reductase in the methanogenic archaeon Methanosarcina acetivorans is mediated by the tfuA-ycaO locus. This experimental procedure showing this is perfect and the results obtained convincing. The excellent manuscript should be published after considering the following minor points:*
*Discussion paragraph three and following: Yes, MCR is generally thought to be the rate limiting enzyme in methanogenesis, however, the only step in methanogenesis that is physiologically irreversible, is the heterodisulfide reductase reaction rather than the MCR reaction. Thus MCR is most probably not rate limiting. This should be considered or at least discussed.*

We have addressed the reviewer’s comment in the Discussion section.

Of the many glycins in McrA only one is converted to a thioglycin and this glycine is buried deep within the McrABC complex. The post-translational modification must therefore occur as long as the protein is not yet folded. Generally enzymes mediating post-translational modifications specifically recognize the sequence upstream and downstream of the amino acid to be modified. Could the authors comment on this in the manuscript with a few sentences?

We have briefly discussed the putative roles of TfuA, including being involved in sequence recognition in the Discussion.

[Editors' note: the author responses to the re-review follow.]

*The consensus is that in the absence of supporting biochemical data, the manuscript still pushes a mechanistic message too strongly. Please moderate the discussion. Your results have shown that the thioglycine within the active site is not essential for full activity, not that it lacks a direct catalytic role. In addition, the paragraph on the discussion of growth phenotypes is convoluted and overly speculative. It would be better to tighten and moderate this paragraph, in the absence of supporting data. Growth phenotypes should also be mentioned in the Abstract, so it is clear that the phenotype of the double mutant extends beyond the lack of a PTM.*

We have modified the Discussion to soften our stance on the mechanistic underpinnings of the thioglycine modification (Discussion, paragraph three). The growth phenotypes are now discussed in the Abstract.

*While we agree that additional biochemical experiments are outside the scope of this work, for this manuscript to be published in eLife as a genetics/physiology paper, these aspects need to be airtight. Towards that end, please include complementation experiments for the mutants.*

As requested by the editor, we have performed complementation experiments for each of the mutants (∆*ycao-tfuA*, ∆*ycaO,* and ∆*tfuA*). Details pertaining to these experiments can be found in the Results section (subsection “The ∆ycaO-tfuA mutant lacks the McrA Gly465 thioamide”) and also in Figure 4 Panels E, F, G.